# Impact Fretting Wear of MoS_2_/C Nanocomposite Coating with Different Carbon Contents under Cycling Low Kinetic Energy

**DOI:** 10.3390/nano11092205

**Published:** 2021-08-27

**Authors:** Junbo Zhou, Lin Zhang, Yuan Ding, Xudong Chen, Zhenbing Cai

**Affiliations:** 1School of Mechanical and Electrical Engineering, Chengdu University of Technology, Chengdu 610059, China; zhoujunbo830@cdut.edu.cn; 2School of Materials and Science, Southwest Jiaotong University, Chengdu 610031, China; 531508439@163.com (Y.D.); chenxd6758@163.com (X.C.); czb_jiaoda@126.com (Z.C.)

**Keywords:** MoS_2_/C coating, impact wear, damage mechanism, dynamic response

## Abstract

MoS_2_/C nanocomposite coatings were deposited on a 304 stainless steel plate by unbalanced magnetron sputtering from carbon and molybdenum disulfide targets, and the target current of MoS_2_ was varied to prepare for coating with different carbon contents. The mechanical and tribological properties of the MoS_2_/C nanocomposite coating with different carbon contents were studied using a low-velocity impact wear machine based on kinetic energy control, and the substrate was used as the comparison material. The atomic content ratio of Mo to S in the MoS_2_/C coating prepared by unbalanced magnetron sputtering was approximately 1.3. The dynamic response and damage analysis revealed that the coating exhibited good impact wear resistance. Under the same experimental conditions, the wear depth of the MoS_2_/C coating was lower than that of the substrate, and the coating exhibited a different dynamic response process as the carbon content increased.

## 1. Introduction

Transition metal dichalcogenides (TMDs) have drawn significant attention in recent years due to their excellent electrical conductivity, good chemical stability, and high mechanical strength [1,2]. The individual crystallites of TMDs have layered structures, which can form a low shear strength friction film in a vacuum or in dry air, resulting in an extremely low coefficient of friction [3]. As a typical representative of TMDs, MoS_2_ coating has been widely used in aerospace and other fields. The main failure modes of coatings in these fields are wear and impact failure, and coatings are difficult to repair in a space environment, so it is necessary to study their friction and impact resistance properties. MoS_2_ coating has good self-lubricating properties that can be attributed to the proper combination of grain size, two-crystal orientation, and absence of contaminants [4,5,6,7]. The MoS_2_ coating under ultra-high-pressure conditions has a coefficient of friction of 0.001 [8,9]. However, the porous columnar structure of pure MoS_2_ coating results in low hardness, low load-carrying capacity, poor film-based bonding properties, and easy tribochemical reaction with air, which limit the practical applications of the MoS_2_ coating [10,11,12,13]. At present, the tribological properties of the MoS_2_ coating are commonly improved by doping other elements or compounds. Doping metal elements, such as Ti, Pb, and Au, can improve the behavior of the MoS_2_ coating, and the oxidation of MoS_2_ can be suppressed by the preferential oxidation of these metals [14,15,16,17,18,19]. Doping metal compounds, such as ZnO, PbO, and TiN, reduce the crystallinity of MoS_2_, but these doped oxides may affect the tribological properties of MoS_2_ [20,21,22]. Doping nonmetallic elements, such as C and N, can improve coating density, hardness, and oxidation resistance, and is an ideal doping method [23,24,25,26,27,28,29,30]. Gu [31] used direct current magnetron sputtering to prepare MoS_2_-C composite films with different carbon contents (carbon content 40.9–73.1 at.%). The amorphous MoS_2_-C coatings structure increased the hardness and improved wear resistance. The coating exhibited a low coefficient of friction (less than 0.1). The composite structure protected MoS_2_ from oxidation and improved the mechanical properties of the coating, which are both excellent friction properties. Pimentel [32] used radio frequency magnetron sputtering to prepare Mo-S-C coatings with different carbon contents and MoS_2_ target power ratios, and the carbon content in the film was between 13 at.% and 55 at.%. The nanocomposite structure embedded in the carbon matrix with molybdenum carbide and molybdenum disulfide particles was characterized by using X-ray diffraction (XRD) and X-ray photoelectron spectroscopy (XPS), and its size was less than a few nanometers. The hardness of the coating increased from 0.7 to 4 GPa with increasing carbon content, and the friction and wear of the Mo-S-C coating in dry nitrogen were negligible. In humid air, the friction was higher at low carbon contents, and the friction generally decreased as the load increased. Xu et al. [33] studied the effects of C doping on the structure, morphology, mechanical properties, and tribological properties of MoS_2_/a-C composite films deposited by using radio frequency magnetron sputtering; when the sputtering power was increased from 0 to 400 W, the hardness of the composite film increased from 0.22 to 2.23 GPa. In the pin-on-disk tests, the film deposited with 300 W of graphite sputtering power exhibited low friction in vacuum and ambient air, but showed different wear and lubrication mechanisms. The Raman spectrum analysis of the tribological films and debris showed that shear strength induced long-range reordering of sputtered MoS_2_ phases into a lamellar structure, whereas most of the carbon content was immediately released from the tribofilm into debris under vacuum sliding conditions. Presently, the tribological properties of MoS_2_/C coatings were mainly tested using the pin-on-disk friction test machine. To further investigate their tribological performance, we used a low-velocity impact wear machine based on kinetic energy control to obtain the dynamic response of the test.

In this study, MoS_2_/C nanocomposite coatings were deposited on a 304 stainless steel plate by using unbalanced magnetron sputtering. The effect of C doping on the structure, morphology, and mechanical behavior of the MoS_2_/C nanocomposite coating was studied. The tribological properties of the MoS_2_/C nanocomposite coating were investigated by using a low-velocity impact wear machine based on kinetic energy control, and the substrate was used as the comparison material. The impact behavior and characteristics of the coatings were determined based on the dynamic response of the energy and wear mechanism in the experiment to determine the optimal carbon content and MoS_2_ content ratio.

## 2. Materials and Methods

A 304 stainless steel plate (30 × 30 mm × 2 mm; Crwt% = 18.0, Niwt% = 9.0) was used as the substrate material, and ultrasonic cleaning was performed in acetone and absolute ethanol for 20 min. MoS_2_/C nanocomposite coatings with different carbon contents were deposited by using an unbalanced magnetron sputtering system (UDP-650). The substrate was placed vertically on the sample holder of a uniaxial rotating structure. A MoS_2_ target of 99.9% purity and a high-purity graphite target were placed opposite each other. Prior to deposition, the vacuum chamber was evacuated to 1 × 10^−3^ Pa. Afterward, the 304 stainless steel plates were sputter-cleaned with Ar^+^ ions for 30 min. A Cr target was used to deposit a transition layer, which improved the adhesion behavior between the coating and the substrate. Finally, the MoS_2_/C composite film was deposited. The working pressure was 1 × 10^−3^ Pa, whereas the DC bias voltage was −50 V. The target currents of MoS_2_ were 0.1, 0.2, 0.3, 0.4, 0.5, 0.6, 0.7, and 0.8 A, whereas the target of C was 3.5 A.

The tribological properties of the coatings were evaluated by using a low-velocity impact wear machine (Figure 1) [34,35]. This device was driven by a voice coil motor and can reciprocate in the positive/cosine mode. The damping punch drove the mass under the excitation of the motor. Owing to the low friction coefficient of the rail (approximately 0.006), the mass could be considered to maintain a constant velocity when impacting and detaching the plane specimen.

The tribological performance of the coating was tested at room temperature. The mass of the impact block was 110 g, and the impact velocity was 50 mm/s. The number of cycles was 10,000. The impactor was a GCr15 steel ball, with a diameter of 9.525 mm. Before the experiment, the flats and impactors were cleaned by using anhydrous ethanol in an ultrasonic bath. The same parameters were tested in the substrate as a reference. Each test was repeated twice.

The surface morphology and cross-section image of the MoS_2_/C composite coating were obtained by using scanning electron microscopy (SEM, JSM-6610, Horiba, Kyoto, Japan), and the elements distribution of the as-deposited coating was detected by using an electron probe micro analyzer (EPMA, JXA-8230, Horiba, Kyoto, Japan). X-ray spectroscopy (EDX, Aztec X-Max 80, Oxford, Abingdon, UK) and XPS (ESCALAB-250Xi, Thermo Fisher Scientific, Waltham, MA, USA) with Al K Alpha radiation were used to obtain the compositions of the coatings. Raman spectroscopy (LabRam HR800, Horiba, Kyoto, Japan) with a 532-nm Ar^+^ laser was used to evaluate the information on the bonding structure. A 2D profiler (NanoMap-500DLS, AEP TECHNOLOGY, San Francisco, CA, USA) and 3D morphology (Contour GT-K1, Bruker, Karlsruhe, GER) were used to measure the scar depth and the damage volume, respectively. Additionally, a nanoindenter (MTS Nanoindenter G200, Agilent, Santa Clara, CA, USA) was used to detect the hardness and the elastic modulus. The maximum indentation depth was 200 nm to minimize the effect of the substrate.

## 3. Results

### 3.1. Coating Deposition and Chemical Composition

The surface morphology and element distribution of MoS_2_0.1 and MoS_2_0.5 are shown in Figure 2. The results were obtained by EPMA. The elemental distributions of C, S, and Mo are even, which indicates that the deposition process was in line with the expectations, and the coating surface composition was consistent.

The chemical composition of the MoS_2_/C films with different deposition conditions are shown in Figure 3. As the MoS_2_ target current increased, the C content in the coating decreased, whereas the Mo and S contents increased. Meanwhile, the S/Mo content ratio was approximately 1.26, which was lower than the theoretical value of two because S was easier to sputter from the surface than Mo, and Ar^+^ erosion reduced the S content in the coating [24]. The marks, MoS_2_0.1–MoS_2_0.8, in the first column refer to the target currents of MoS_2_ during sputtering.

The cross-section morphology of the coatings is shown in Figure 4. The thickness of MoS_2_0.1 was approximately 0.91 μm, while the thickness of MoS_2_0.8 was approximately 1.41 μm. As the MoS_2_ target current increased, the thickness of the coating thickened. To determine the mechanical properties of the coating, the coating should not be destroyed during the impact test. Therefore, we selected a light mass and a small impact velocity.

The hardness and elastic modulus of the MoS_2_/C coatings are shown in Figure 5. As the MoS_2_ target current increased, the hardness and elastic modulus of the coating increased first and then decreased and reached the maximum when the target current of MoS_2_ was 0.3 A. The hardness of the pure MoS_2_ coating was 2.6 GPa and the elastic modulus was 48 Gpa [36], which were much lower than those of the MoS_2_/C coatings. The hardness of the coatings was higher than that of the substrate, which means that the MoS_2_/C coatings exhibited good mechanical properties. The load–displacement curve indicates that as the MoS_2_ target current increased, the elastic recovery of the composite film decreased linearly, which may refine the film load-bearing capacity in the same way.

XPS analysis was performed to explore the chemical composition of the MoS_2_/C coating. Figure 6 shows the XPS spectra of the MoS_2_/C0.5 films. The C 1s, S 2p, and Mo 3d spectra of MoS_2_0.5 were fitted by the Guassian–Lorentian function to analyze the chemical forms. In the S 2p and Mo 3d spectra, the intensity decreased with carbon content. Figure 6b shows the C 1s XPS spectrum of the composite coatings with different C contents. The C 1s spectrum was fitted into three components, representing the C-C bond (287.0 eV), the C=C bond (284.5 eV), and the molybdenum carbide (283.6 eV) [37]. Figure 6c shows the S 2p XPS spectra of the coatings with different carbon contents, which were fitted into four components, representing the MoS_2_ structure (163.2 eV and 162.1 eV) and MoS_2_ structure (161.8 eV and 160.6 eV) [18]. In Figure 6d, the peaks are observed in the Mo-S (227.5 eV and 230.69 eV) and Mo-O (231.3 eV and 234.6 eV) structures. The Mo 3d spectrum of all samples showed a small shoulder at approximately 226.9 eV, which was represented by the Mo 3d peak [38,39]. The Mo-C bond was confirmed in the Mo 3d spectra at 227.9 and 231.0 eV, which was consistent with the analysis of the C 1s spectrum [32]. The appearance of Mo-O indicated that the surface of the coating was oxidized. The Mo-C and Mo-S peak intensities in the film and the oxygen content increased with the MoS_2_ content.

To further investigate the structure of the composite coatings, Raman spectra are presented in Figure 7. The Raman spectra can be divided into three parts, namely, MoS_2_ (250–500 cm^−1^), MoO_3_ (750–1000 cm^−1^), and C (1000–1700 cm^−1^) [24]. Two major peaks are observed (approx. at 1396 and 1510 cm^−1^) corresponding to carbon, and peaks in MoS_2_ (260 cm^−1^) and MoO_3_ (910 cm^−1^) are also observed. The relatively sharp peaks of MoS_2_ in the deposited film can be attributed to thermal crystallization through its high laser energy during the Raman test [31]; the sharp peaks disappear when a low laser energy is used. The intensity of the MoS_2_ peaks increased with the MoS_2_ target power, which is consistent with the XPS results. According to the ID/IG, as the content of C element decreases, the defects of C element crystal lessen, so it is recommended to strictly control the target current of C when preparing MoS_2_/C by this method.

### 3.2. Dynamic Response of Impact Wear Test

Coatings generally have superior mechanical and tribological properties to those of substrates. In theory, increasing the damage to the material surface will change the dynamic response. Therefore, the damage of the surface can be estimated based on the dynamic response of the coating and the substrate during the experiment. The effect of C doping on the mechanical and tribological properties of MoS_2_/C composite coating was investigated using a low-velocity impact wear machine.

Figure 8 shows the evolution of impact force for different material parameters. As another form of dynamic response in impact testing, the waveform of the impact force also changes with different materials. The peak force of MoS_2_/C is higher than that of the substrate, indicating that the hardness of the coating was higher. The impact forces at different cycle times are shown in Figure 8a–c. The impact force of the coatings is higher than that of the substrates. The peak impact force of MoS_2_0.4 is always the largest, while that of the MoS_2_0.6 is the smallest. Furthermore, the contact time of the MoS_2_ coating is lower than that of the substrate as expected. Figure 8d shows that the evolution of peak impact force has three stages. In stage I, the impact force decreases because the wear scar increases with the cycles of impacts, resulting in an increase in the contact area of the ball with the plate. In stage II, the impact force remains basically the same because the wear scar has reached the maximum at this time, and the contact area no longer increases. In stage III, the impact force increases because work hardening occurs.

In addition to the kinetic energy and deformation during impact, other forms of energy conversion exist, such as frictional heat and crack propagation. In the case of low-speed impacts, the part of the thermal energy is generally negligible. Accordingly, this study assumed that the initial impact energy only transformed into deformation energy, kinetic energy, and the energy consumed in materials degradation. Therefore, the energy change of the sample can be analyzed by the velocity curve of the impact test.

Figure 9 shows the dynamics velocity response under different cycles. The initial impact velocity was 50 mm/s, and the difference in rebound velocity reflects the difference in energy absorption. The rebound velocity of the coatings first decreased and then increased with the decreasing carbon content. The trend of the rebounded velocity was consistent at both 100 and 10,000 cycles, but ΔV gradually increased, because the wear scar gradually deepened as the number of impacts increased, and work hardening occurred at the contact position.

Using the same materials, the lower the rebound speed was, the more energy the sample would absorb. Figure 10 shows the curve of the kinetic energy during the test; the kinetic energy is calculated by a simple formula, E = 1/2 mv^2^, where v is the data from Figure 9, and m is the mass of the impact block, which is 110 g. Therefore, the initial impact kinetic energy is 0.14 (±5%) mJ. The absorbed energy of the MoS_2_/C coatings was higher than that of the substrate. The absorbed energy first decreased and then increased with the decreasing carbon content, which indicates that MoS_2_0.3 absorbs the most energy, while MoS_2_0.6 absorbs the least energy.

Figure 11a shows the energy distribution and energy absorption rate of different material parameters. Total energy is the sum of the initial kinetic energies in 10^4^ impact experiments. Remaining energy is the sum of the returning kinetic energies in 10,000 impact experiments. Absorbed energy is the difference between total energy and remaining energy. The energy absorption rate of the coating was approximately 0.65–0.78 higher than that of the substrate. The results show that the MoS_2_/C coating absorbed more energy than the substrate under the same experimental conditions. The specific causes of this phenomenon are discussed in Section 3.3.

The evolution of the energy absorption ratio of MoS_2_/C and its substrates is shown in Figure 11b. The energy absorption ratio of MoS_2_/C slowly decreased and then stabilized at approximately the 2000th cycle. At the beginning of the test, the contact between the impact ball and the plane sample could be regarded as a point contact. At this time, the contact area was small, resulting in a high contact stress between the interfaces and increasing the susceptibility of the sample to plastic deformation and material removal. Therefore, in this process, the coating and the substrate exhibited a high level of energy absorption ratio. As the test progressed and material damage accumulated, the contact area increased and resulted in a decrease in contact stress. In addition, the surface of the wear scar resulted in work hardening during the impact test, which increased the surface strength and improved the wear resistance. Figure 12 shows the deformation of different coatings and substrates, which was obtained by integrating velocity. Consequently, the deformation gradually increased.

### 3.3. Impact Wear Analysis

The 3D profile of impact damage was observed through the 3D morphology, and the wear volume was calculated. The results are shown in Figure 13. The wear of the substrate was serious, and severe deformation and accumulation had occurred. Meanwhile, a large amount of abrasive debris had accumulated on the surface of the wear scar. The wear scar area of MoS_2_0.1 was relatively large, but the depth was shallow. The damage of MoS_2_0.3 was worse than that of MoS_2_0.1, which indicates that the wear scar area was similar but that of MoS_2_0.3 was deeper. MoS_2_0.4 had a larger and deeper wear scar area, while MoS_2_0.6 had a lower wear scar area with the slightest wear volume than those of the other samples. The wear of the coating was consistent with the dynamic response in the experiment. Impact damage mainly consisted of plastic deformation and interface wear, depending on the mechanical properties of the material. Under the same experimental conditions, MoS_2_0.4 absorbed more energy and had serious damage, while MoS_2_0.6 absorbed less energy and had less damage, demonstrating that MoS_2_0.6 has a better performance. In Figure 11, the energy absorption ratio of the substrate is lower than that of the MoS_2_/C coatings, but the wear damage is more serious. This shows that during the impact process, the coating absorbs more energy with little damage, and their mechanical properties are better than the substrate. Figure 13j shows the cross-sectional profiles of all the samples. After 10^4^ cycles of impact test, the wear volume of the substrate is larger than those of the coatings.

Figure 14a shows the maximum wear depth of the samples after the impact test. The substrate had a maximum wear depth of 1.082 μm, while the coatings had a smaller wear depth. Figure 14b shows the wear volume of all the samples. MoS_2_0.4 had the largest wear volume of approximately 9937 μm^3^, and MoS_2_0.6 had the smallest wear volume of approximately 3117 μm^3^. Except for MoS_2_0.4, all other coatings had a smaller wear volume than the substrate. Thus, MoS_2_0.6 has a good impact resistance and can provide good protection for its substrate.

To determine the wear mechanism during the impact process, SEM and EDX were used to observe the wear scar and measure the element distribution. Figure 15 shows the SEM images and EDX of the substrate; a significant peeling of the material in the upper part of the worn area is observed. According to the analysis results of EDX (point 1 is the distribution of substrate material elements), the C and Si contents of point 2 increased greatly, while the Fe element content decreased, so point 2 should be sandstone impurities; at point 3, the C and Fe contents decreased, and the O content increased greatly. Meanwhile, point 3 was in the crack area, so the material at point 3 should be frictionally oxidized, and the material failed and cracked; the element distribution at point 4 was similar to that at point 3, so point 4 should be the accumulation of material after frictional oxidation. In summary, the substrate underwent friction oxidation during the impact process, causing the oxide film to peel and aggravating the material loss. Figure 16 shows the SEM images and EDX of MoS_2_0.4. The wear volume of MoS_2_0.4 was the largest, so elemental distribution analysis was carried out to explore its wear mechanism. The wear scar of MoS_2_0.4 was relatively flat, with no obvious cracks or defects. The analysis result shows that the element contents in the two regions were basically the same. Plastic deformation occurred on the surface of the material during the entire impact process and no obvious chemical reaction occurred. The SEM images and EDX of MoS_2_0.6 are shown in Figure 17. Compared with the element distribution of point 1, the oxygen content in the area where the wear debris accumulated (point 2) was greatly increased, while the element contents of C, S, and Mo decreased, demonstrating that frictional oxidation probably occurred. Point 3 shows a different surface morphology than point 1. Analysis shows that the composition was the same as that in point 1 because of the fault left by the surface peeling of the material. Compared with point 1, the oxygen and carbon contents of point 4 increased, and the S and Mo contents decreased, which should have been the accumulation of materials.

## 4. Conclusions

MoS_2_/C coatings with different carbon contents (69 at.%–84 at.%) were prepared by unbalanced magnetron sputtering. The structure, mechanical properties, and tribological properties of MoS_2_/C coatings with different carbon contents were investigated. The impact wear performance was analyzed using a low-velocity impact wear machine. The following conclusions were drawn:(1)The distribution of the C, S, and Mo elements in the prepared MoS_2_/C coating was uniform, indicating that the surface properties of the composite coating were consistent. Meanwhile, the S/Mo content ratio was approximately 1.26. C doping increased the hardness and oxidation resistance of the MoS_2_ coating.(2)C doping increased the hardness of the coatings (6.4–12 Gpa), which further improved the wear resistance of the MoS_2_/C coatings.(3)When the target current of C was 3.5 A and the target current of MoS_2_ was 0.6 A (the carbon content was approximately 78.3%), the coating exhibited a good impact resistance in terms of dynamic response, including low energy absorption ratio and low impact force.(4)The wear scar and the volume loss analysis show that MoS_2_0.6 has a good impact resistance and can protect the substrate. The wear mechanism of coatings presents fatigue delamination and spalling, which are caused by cyclic shear stress. In addition, frictional oxidation occurs, especially in the stratified zone.

## Figures and Tables

**Figure 1 nanomaterials-11-02205-f001:**
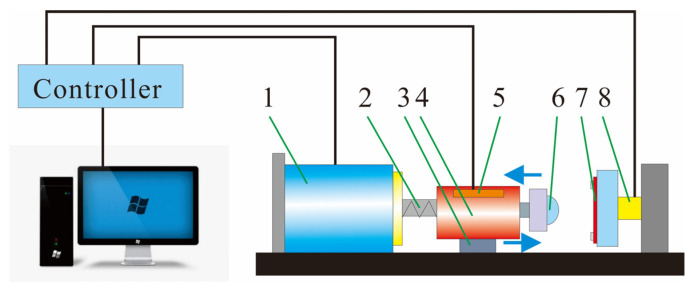
Low-velocity impact wear machine: (1) voice coil motor, (2) damping punch, (3) rail, (4) impact block, (5) motion detector, (6) impact ball, (7) sample, (8) force sensor.

**Figure 2 nanomaterials-11-02205-f002:**
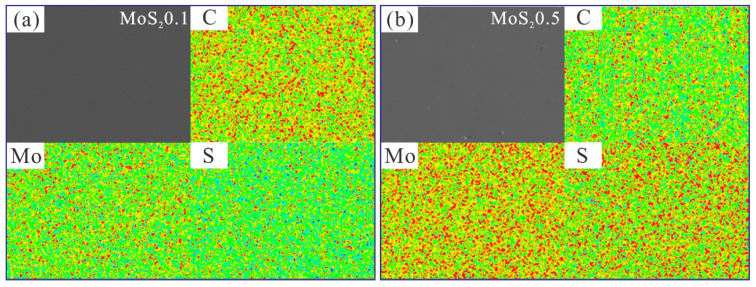
SEM micrographs and element distribution of MoS_2_/C composite coating. (**a**) MoS_2_0.1, (**b**) MoS_2_0.5.

**Figure 3 nanomaterials-11-02205-f003:**
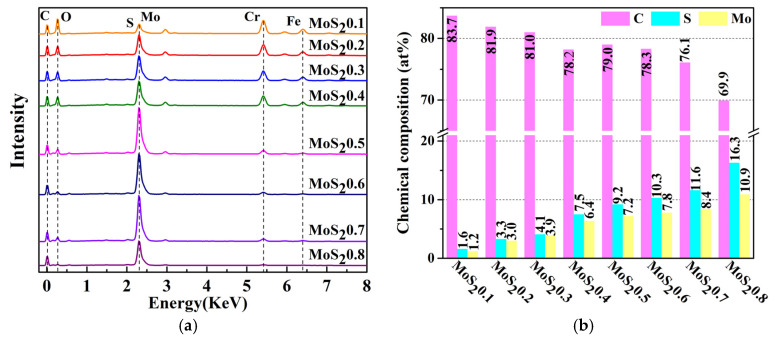
Chemical composition of the MoS_2_/C films. (**a**) EDX and (**b**) Chemical composition of EDX.

**Figure 4 nanomaterials-11-02205-f004:**
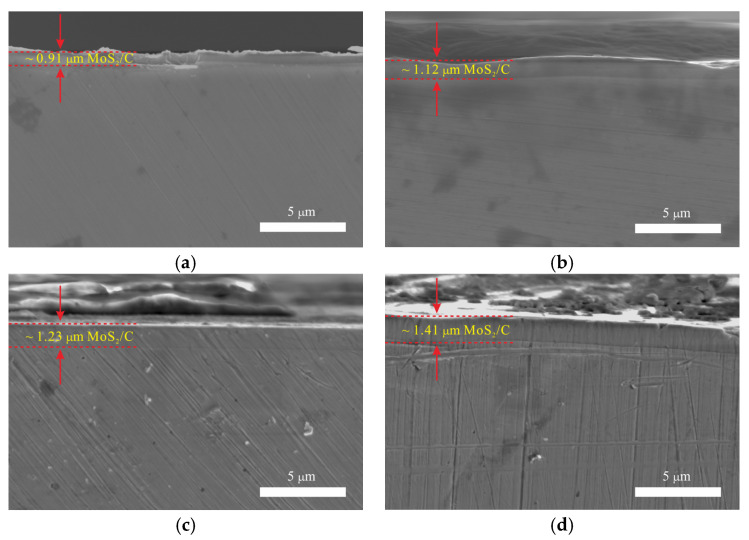
Cross-section morphology of MoS_2_/C composite coating. (**a**) Cross-section of MoS_2_0.1, (**b**) Cross-section of MoS_2_0.5, (**c**) Cross-section of MoS_2_0.6 and (**d**) Cross-section of MoS_2_0.8.

**Figure 5 nanomaterials-11-02205-f005:**
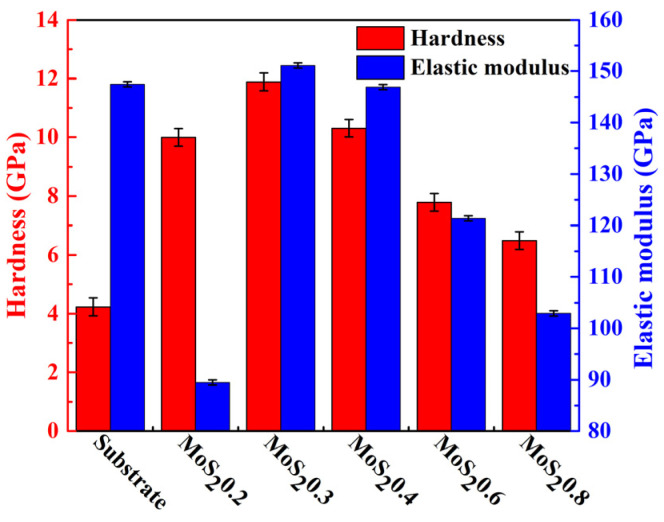
Hardness and elastic modulus of the MoS_2_/C coatings.

**Figure 6 nanomaterials-11-02205-f006:**
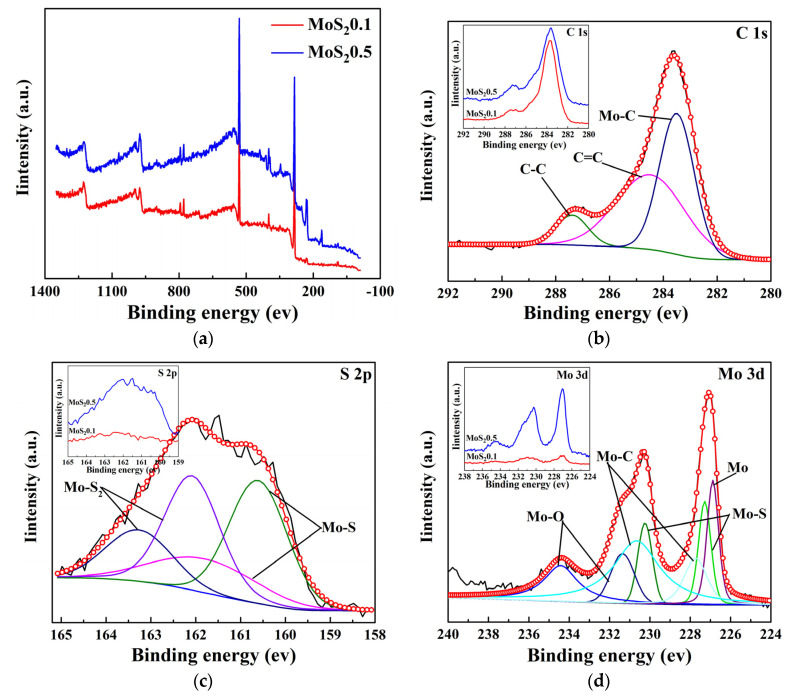
XPS spectra of MoS2/C composite coating. (**a**) XPS spectra, (**b**) C 1s of MoS_2_0.5, (**c**) S 2p of MoS_2_0.5 and (**d**) Mo 3d of MoS_2_0.5.

**Figure 7 nanomaterials-11-02205-f007:**
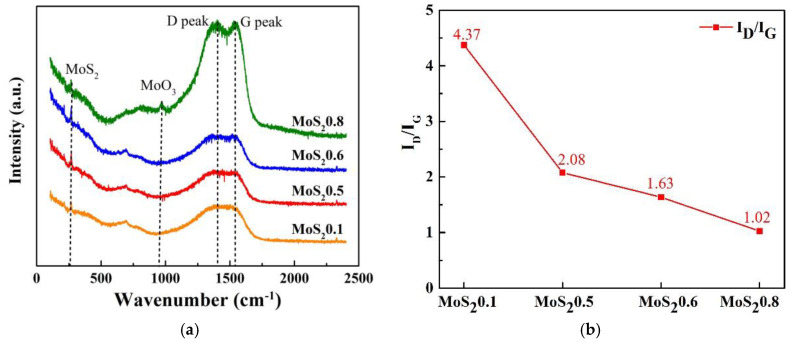
Raman spectra of MoS_2_/C composite coating. (**a**) Raman spectra, (**b**) ID/IG.

**Figure 8 nanomaterials-11-02205-f008:**
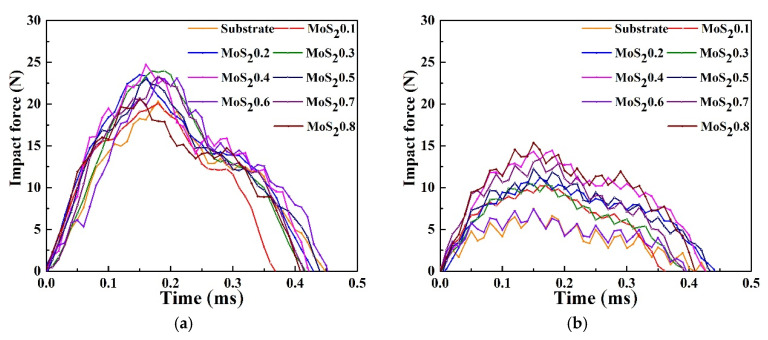
Impact force under different cycles. (**a**) Cycle = 10^2^, (**b**) Cycle = 10^3^, (**c**) Cycle = 10^4^ and (**d**) Evolution of peak impact force.

**Figure 9 nanomaterials-11-02205-f009:**
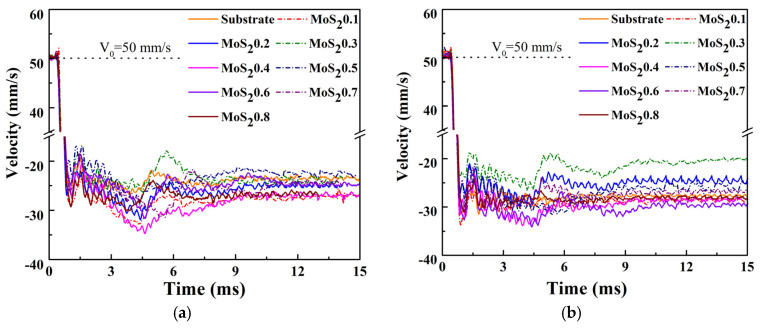
Dynamics velocity response under different cycles. (**a**) Cycle = 10^2^, (**b**) Cycle = 10^4^, (**c**) ΔV in different cycles.

**Figure 10 nanomaterials-11-02205-f010:**
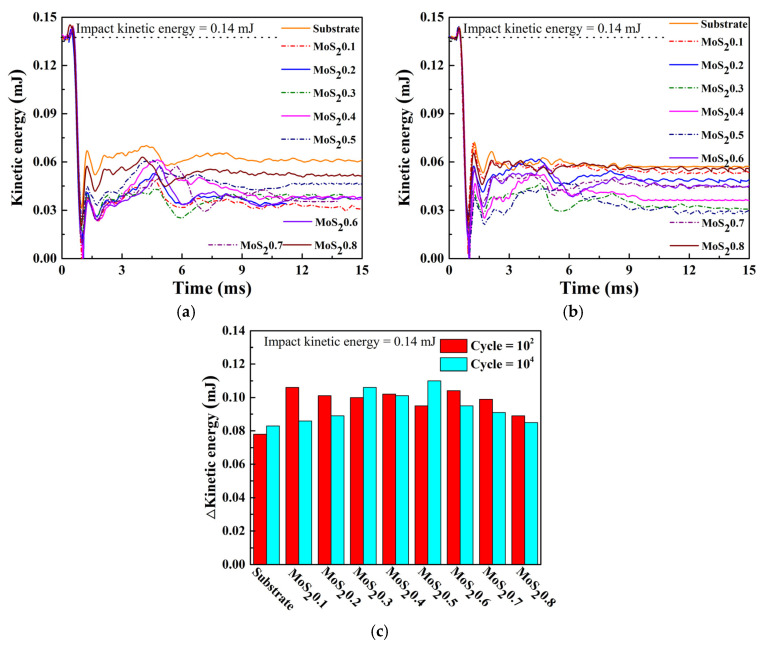
Change in kinetic energy during varied impact cycles. (**a**) Cycle = 10^2^, (**b**) Cycle = 10^4^, (**c**) ΔE in different cycles.

**Figure 11 nanomaterials-11-02205-f011:**
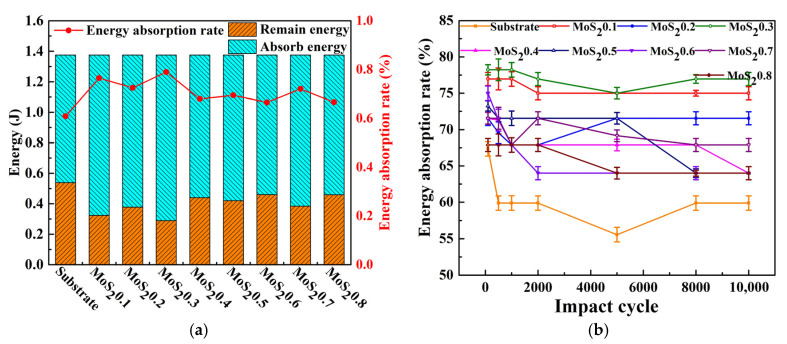
Energy distribution and energy absorption ratio. (**a**) Total energy and its distribution and (**b**) Evolution of energy ratio.

**Figure 12 nanomaterials-11-02205-f012:**
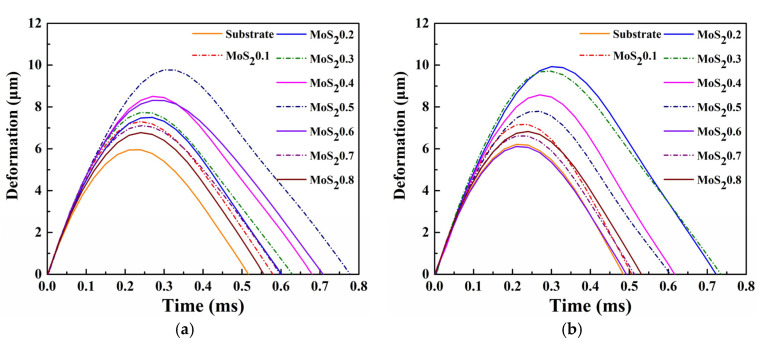
Deformation of different coatings and substrates. (**a**) Cycle = 10^2^ and (**b**) Cycle = 10^4^.

**Figure 13 nanomaterials-11-02205-f013:**
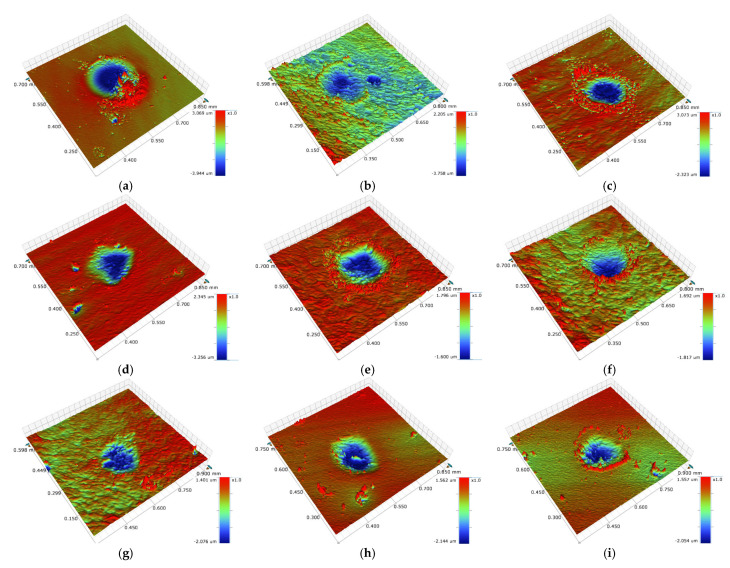
Three-dimensional morphology and cross-sectional profiles. (**a**) Substrate, (**b**) MoS_2_0.1, (**c**) MoS_2_0.2, (**d**) MoS_2_0.3, (**e**) MoS_2_0.4, (**f**) MoS_2_0.5, (**g**) MoS_2_0.6, (**h**) MoS_2_0.7, (**i**) MoS_2_0.8, (**j**) Cross-sectional profiles.

**Figure 14 nanomaterials-11-02205-f014:**
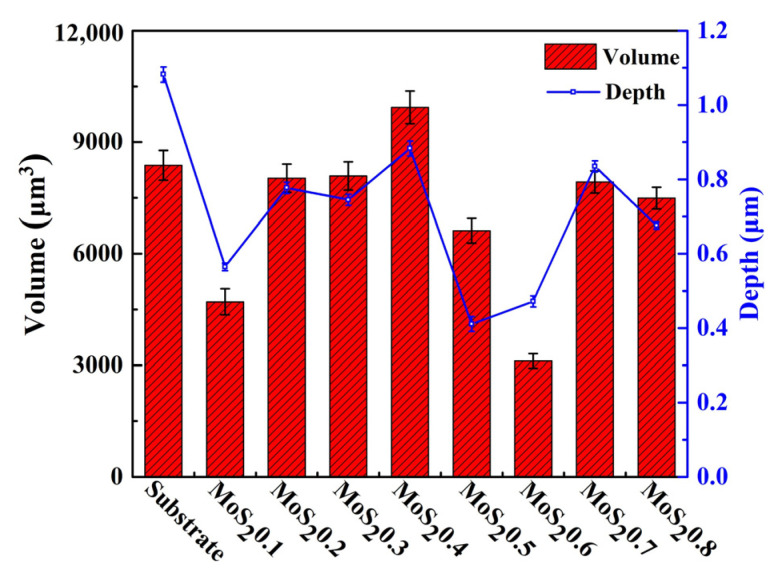
Maximum wear depth and wear volume.

**Figure 15 nanomaterials-11-02205-f015:**
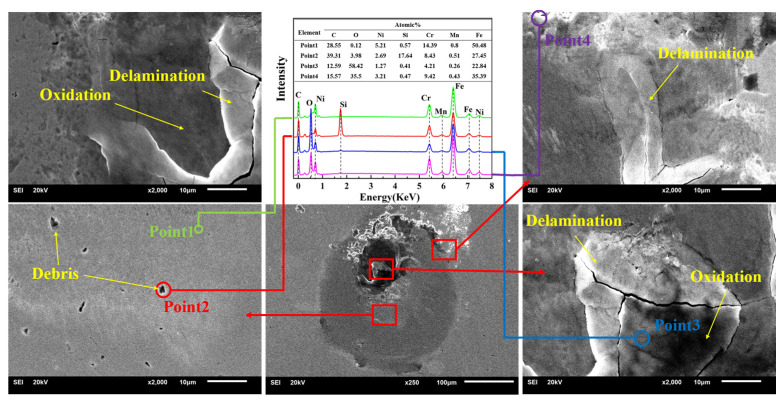
SEM images and EDX of the substrate.

**Figure 16 nanomaterials-11-02205-f016:**
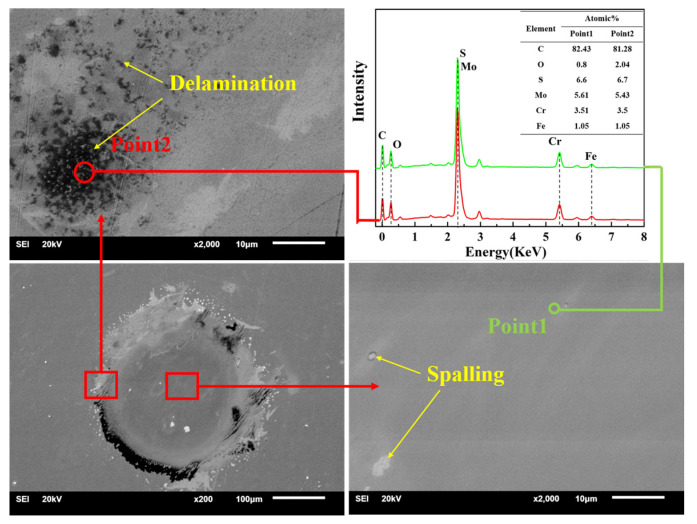
SEM images and EDX of MoS_2_0.4.

**Figure 17 nanomaterials-11-02205-f017:**
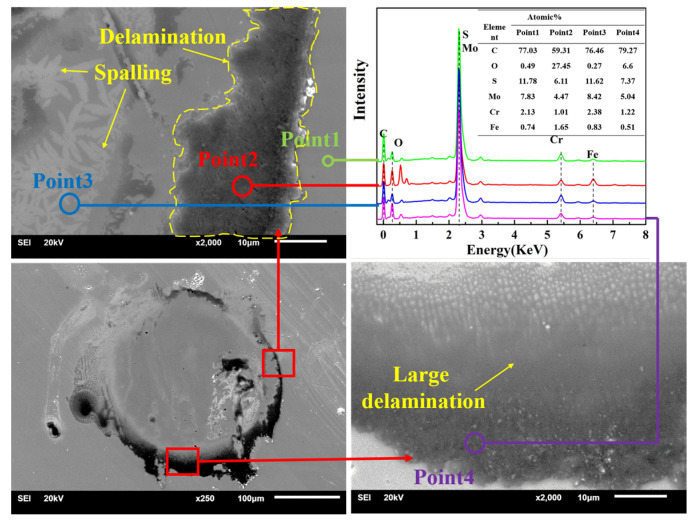
SEM images and EDX of MoS_2_0.6.

## Data Availability

Data sharing is not applicable to this article.

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
