# Peer review of "Impact Fretting Wear of MoS2/C Nanocomposite Coating with Different Carbon Contents under Cycling Low Kinetic Energy"

_nanomaterials, 2021, doi:10.3390/nano11092205_

Round 1

Reviewer 1 Report

The manuscript describes the results of an experimental study into the tribological behaviour of a composite MoS2/C coating subjected to cyclic impacts. The study is based on a solid methodology and the results are sound. The manuscript is well organized. It requires an English check to correct several style and Grammar errors.

I have only a few comments/questions:

1) identification of plots on the figures is difficult because of the use of similar colours.

2) What is the motivation to study the coatings under impact behaviour? This can be explained in the Introduction section.

Author Response

  • identification of plots on the figures is difficult because of the use of similar colours.

Answer: Some adjustments were made to all images

2) What is the motivation to study the coatings under impact behaviour? This can be explained in the Introduction section.

Answer:As a typical representative of TMDs, MoS2 coating has been widely used in aerospace and other fields, the main failure modes of coatings in these fields are wear and impact failure, and coatings are difficult to repair in the space environment, so it is necessary to study their friction and impact resistance properties.(Line 26)

Reviewer 2 Report

Dear Authors,

The article describes deposition technology and examined the chemical and phase composition, mechanical properties and the results of tribological tests using a low-velocity impact wear machine MoS2/C coatings produced by the unbalanced magnetron sputtering method. The research methodology was described by the authors in their earlier works [34, 35].

In the introduction, there is no information for what purpose these coatings were deposited and what their potential application may be.

Here are some comments and suggestions:

Line 80: No standard information available regarding the structure and chemical composition of 304 steel

Line 86: Error in noting pressure unit in Pascals (Pa) - should be lowercase "a".

Lines 87-88: What purity was the Cr target used? What deposition parameters were used and thickness of the Cr transition layer.

Line105: No standard for GCe15 bearing steel and what properties this steel?

Lines 118-119 and Fig. 5: Using an indenter penetration depth of 200 nm is well above 10% of the coating thickness, so the substrate will influence the hardness measurements of the coatings.

Line 140 and Fig. 4: Did the tested coatings differ in thickness? If so, it is difficult to compare the mechanical and tribological properties.

Line 147: The hardness units have spelling error (GPa).

Line 148: Should be plural - "coatings".

Chapter 3.1: Chemical composition and structure studies indicate that these were nanostructured carbon coatings (DLC) doped with MoS2 - no such term is available.

Line 243: Value v not read from Fig. 3?

Line 300: It should read "Fig. 13(j)"

Lines 302-310, Figure 13: No information on how the counter-samples performed. Were such observations and analyzes carried out?

Lines 316-317 and 367-379: MoS2 0.6 coatings have been found to have a good impact resistance and can provide good protection for its substrate. No attempt has been made to explain where this comes from.

Line 433: Error in writing - it should read "MoS2 / C"

The article is acceptable after introducing minor corrections and additions.

Greetings

Author Response

Line 80: No standard information available regarding the structure and chemical composition of 304 steel

Answer: 304 steel strictly comply with ASTM A276/A276M-15 standard, we added its Cr, Ni element content in the text.

Line 86: Error in noting pressure unit in Pascals (Pa) - should be lowercase "a".

Answer: it has been corrected.

Lines 87-88: What purity was the Cr target used? What deposition parameters were used and thickness of the Cr transition layer.

Answer: The purity of Cr target is 99.95%, The deposition time is 2min and the deposition thickness is less than 8mm.

Line105: No standard for GCr15 bearing steel and what properties this steel?

Answer: GCr15 steel strictly comply with ASTM A295:1998 standard, which has high hardness and good wear resistance, the steel often used as a bearing steel.

Lines 118-119 and Fig. 5: Using an indenter penetration depth of 200 nm is well above 10% of the coating thickness, so the substrate will influence the hardness measurements of the coatings.

Answer: the substrate will indeed influence the hardness of the coatings. But it does not affect our conclusions, because all coatings are processed with the same criteria and their influence by the substrate is considered the same

Line 140 and Fig. 4: Did the tested coatings differ in thickness? If so, it is difficult to compare the mechanical and tribological properties.

Answer: As the MoS2 target current increases, the thickness of the coating thickens. However, it is a difficult process to ensure the same thickness, more research is needed in the subsequent work.

Line 147: The hardness units have spelling error (GPa).

Answer: it has been corrected.

Line 148: Should be plural - "coatings".

Answer: it has been corrected.

Chapter 3.1: Chemical composition and structure studies indicate that these were nanostructured carbon coatings (DLC) doped with MoS2 - no such term is available.

Answer: it has been corrected.

Line 243: Value v not read from Fig. 3?

Answer: v is the data from Fig. 9

Line 300: It should read "Fig. 13(j)"

Answer: it has been corrected.

Lines 302-310, Figure 13: No information on how the counter-samples performed. Were such observations and analyzes carried out?

Answer:Fig. 15 shows the SEM images and EDX of the substrate; a significant peeling of the material in the upper part of the worn area is observed. In summary, the substrate undergoes friction oxidation during the impact process, causing the oxide film to peel and aggravating the material loss. The existing characterizations are thought to provide a complete characterization of the impact wear mechanism, so no more analyzes on the counter-samples

Lines 316-317 and 367-379: MoS2 0.6 coatings have been found to have a good impact resistance and can provide good protection for its substrate. No attempt has been made to explain where this comes from.

Answer: Fig. 14 (a) shows the maximum wear depth of the samples after the impact test. The substrate has a maximum wear depth of 1.082 μm, while the coatings has a smaller wear depth. Fig. 14 (b) shows the wear volume of all the samples. MoS2.04 has the largest wear volume of approximately 9937 μm3, and MoS20.6 has the smallest wear volume of ap-proximately 3117 μm3. Except for MoS20.4, all other coatings have a smaller wear volume than the substrate. Thus, MoS20.6 has a good impact resistance and can provide good protection for its substrate.

Line 433: Error in writing - it should read "MoS2 / C"

Answer: it has been corrected.